# Biomechanical Study of Symmetric Bending and Lifting Behavior in Weightlifter with Lumbar L4-L5 Disc Herniation and Physiological Straightening Using Finite Element Simulation

**DOI:** 10.3390/bioengineering11080825

**Published:** 2024-08-12

**Authors:** Caiting Zhang, Yang Song, Qiaolin Zhang, Ee-Chon Teo, Wei Liu

**Affiliations:** 1Faculty of Sports Science, Ningbo University, Ningbo 315211, China; zhangcaiting_nbu@foxmail.com (C.Z.); zhang.qiaolin@phd.uni-obuda.hu (Q.Z.); eechon-teo@nbu.edu.cn (E.-C.T.); 2Department of Biomedical Engineering, Faculty of Engineering, The Hong Kong Polytechnic University, Hong Kong 999077, China; 3Doctoral School on Safety and Security Science, Óbuda University, 1034 Budapest, Hungary; 4Faculty of Engineering, University of Szeged, 6724 Szeged, Hungary

**Keywords:** finite element, physiological straightening, herniated disc, bending and lifting, biomechanical

## Abstract

Background: Physiological curvature changes of the lumbar spine and disc herniation can cause abnormal biomechanical responses of the lumbar spine. Finite element (FE) studies on special weightlifter models are limited, yet understanding stress in damaged lumbar spines is crucial for preventing and rehabilitating lumbar diseases. This study analyzes the biomechanical responses of a weightlifter with lumbar straightening and L4-L5 disc herniation during symmetric bending and lifting to optimize training and rehabilitation. Methods: Based on the weightlifter’s computed tomography (CT) data, an FE lumbar spine model (L1-L5) was established. The model included normal intervertebral discs (IVDs), vertebral endplates, ligaments, and a degenerated L4-L5 disc. The bending angle was set to 45°, and weights of 15 kg, 20 kg, and 25 kg were used. The flexion moment for lifting these weights was theoretically calculated. The model was tilted at 45° in Abaqus 2021 (Dassault Systèmes Simulia Corp., Johnston, RI, USA), with L5 constrained in all six degrees of freedom. A vertical load equivalent to the weightlifter’s body mass and the calculated flexion moments were applied to L1 to simulate the weightlifter’s bending and lifting behavior. Biomechanical responses within the lumbar spine were then analyzed. Results: The displacement and range of motion (ROM) of the lumbar spine were similar under all three loading conditions. The flexion degree increased with the load, while extension remained unchanged. Right-side movement and bending showed minimal change, with slightly more right rotation. Stress distribution trends were similar across loads, primarily concentrated in the vertebral body, increasing with load. Maximum stress occurred at the anterior inferior margin of L5, with significant stress at the posterior joints, ligaments, and spinous processes. The posterior L5 and margins of L1 and L5 experienced high stress. The degenerated L4-L5 IVD showed stress concentration on its edges, with significant stress also on L3-L4 IVD. Stress distribution in the lumbar spine was uneven. Conclusions: Our findings highlight the impact on spinal biomechanics and suggest reducing anisotropic loading and being cautious of loaded flexion positions affecting posterior joints, IVDs, and vertebrae. This study offers valuable insights for the rehabilitation and treatment of similar patients.

## 1. Introduction

Repetitive loading due to body posture, such as bending, is well documented as a contributor to lumbar spine pathologies, including intervertebral disc (IVD) issues and changes in physiological curvature. IVDs function as shock absorbers, enabling flexibility for bending and rotation. The natural lordotic curvature of the lumbar spine aids in evenly distributing mechanical loads during activities like standing and lifting. This curvature alleviates pressure on the IVD and lumbar vertebrae and enhances spinal flexibility and stability [1,2]. Athletes confront gravitational forces and the imperative of maintaining mechanical equilibrium within the athletic realm, particularly in weightlifting. This results in the spinal column enduring substantial axial loads and shear stresses, significantly contributing to the prevalence of lumbar spine disorders among sports-related injuries, with a reported prevalence exceeding 80% [3,4]. Despite stringent health protocols in their training, weightlifters still exceed physiological stress thresholds due to intensifying competition and athletes’ relentless pursuit of excellence. Unfortunately, such circumstances often lead to chronic lumbar spine conditions, of which IVD herniation is a prototypical manifestation [5,6]. When the human disc herniates and pathological changes occur, this usually leads to lower back pain and degenerative changes or inflammation caused by cumulative training in the athlete’s role [7]. As a result, the original structure of the nucleus pulposus and annulus fibularis is abnormal, and the original function of load distribution and buffer is destroyed, resulting in uneven stress distribution and anisotropy during activity [8] and, because of its structural failure and bringing other skeletal structural abnormalities of the chain reaction, resulting in movement and daily life in the ability to perform the behavior of the spine instability, including small joints and cartilage tissue, ligaments, and so on, the force and its functional properties change [9,10,11]. Furthermore, prolonged training over the years can impact the intrinsic compensatory mechanisms, potentially influencing the curvature of the lumbar sagittal profile [12]. It is noteworthy, however, that individual genetic predispositions constrain these mechanisms. Concurrently, the anatomical characteristics of the spine are susceptible to alterations stemming from discopathy, with a pronounced interrelationship between the two phenomena [13,14,15,16,17]. This condition introduces several biomechanical deficiencies, culminating in aberrant body postures during mechanical activity, constrained balance capabilities, and restricted individual joint mobility [18].

Manual material handling is a common labor practice in daily life, with tasks such as bending to lift objects posing a high risk for lumbar spine injury [19]. This behavior is prevalent among both professional workers, such as gardeners and movers, and non-professionals engaging in activities like relocating or moving household items [20]. For athletes, their training often includes exercises targeting arm strength, body balance, and localized muscle groups, such as kettlebell or barbell training. Unlike weightlifters who engage in high-load training, manual labor typically involves lighter loads and less stringent body posture requirements. However, performing such activities on already diseased lumbar spines appears to be a random behavior. Hence, exploring the biomechanical response of the lesion lumbar spine to manual material handling is crucial. This relates to athletes’ training and competitions and affects their retirement careers and lumbar spine rehabilitation.

Understanding the internal loads within IVD is crucial for preventing injuries during exercise and activities. Investigating the load distribution in specific areas of the human body often requires considering the overall response. Therefore, finite element (FE) modeling is widely utilized in biomechanical studies of the lumbar spine. FE models of individual segments, including lumbar vertebrae L1-L5, have been developed [21,22,23,24,25]. These models are utilized to investigate material properties, surgical interventions, biomechanical effects of lumbar loads, and boundary conditions, aiming to optimize treatment methods and reduce postoperative recurrence rates. Multiple reports have been previously published on establishing FE models for specific healthy lumbar spine weightlifters to examine their biomechanical responses [26,27,28]. Analyzing internal stresses under various conditions to identify factors contributing to injury risk and offer guidance for preventing and treating sports injuries. In addition, some studies have focused on IVD herniation symptoms. For instance, Mimura et al. [29] explored the relationship between lumbar IVD herniation and reduced multidirectional spine flexibility, finding that affected lumbar discs had reduced range of motion (ROM) in flexion and lateral bending motion under different load conditions. Raheem et al. [30] used a three-dimensional FE model to investigate how changes in the status of the nucleus pulposus lead to lumbar IVD herniation, highlighting significant changes in disc response after nucleus pulposus removal. Yu et al. [31] developed and validated a high-precision three-dimensional FE model of the L4-L5 lumbar spine segment, encompassing both normal and various degrees of IVD herniation, and simulated the mechanical performance of these models under different conditions to mimic human physiological activities. While these reports offer diverse and in-depth perspectives, there is limited focus on specific weightlifter considerations.

This study established a personalized three-dimensional FE model of the weightlifter with physiological straightening and lumbar IVD herniation and analyzed the internal stress changes in the damaged lumbar vertebrae when simulating symmetric bending and weightlifting movements. It was hypothesized that mechanical loading would cause uneven stress distribution in the weightlifter’s lumbar spine and present abnormal mechanical transmission. These results will provide a scientific basis for daily preventive measures, physical interventions, and clinical surgical methods for similar patients.

## 2. Materials and Methods

### 2.1. Participant

The subject of this investigation is a professional weightlifter, aged 21 years, with a height of 169 cm and a body weight of 71 kg. A seasoned physician clinically diagnosed the weightlifter with an L4-L5 IVD herniation, characterized by nucleus pulposus protrusion into the vertebral canal, exerting pressure on adjacent nerve roots, and evaluated the lumbar segmental curvature using the Cobb angle [32,33]. In the computed tomography (CT) image, to measure the Cobb angle, a horizontal line is drawn at the upper edge of the upper vertebra and another at the lower edge of the lower vertebra. Perpendicular lines are then drawn from each of these horizontal lines, and the angle between these perpendicular lines represents the Cobb angle, as shown in Figure 1. A comprehensive assessment of symptomatic manifestations, including pronounced lower back and leg pain, ambulatory difficulties, and disturbances in sensory perception, confirmed the diagnosis. Informed consent, emphasizing the voluntary nature of participation, was obtained in writing from the participant. The Ethics Committee of Ningbo University granted ethical approval for this study, verifying its adherence to established ethical standards (protocol code: RAGH 20240118).

### 2.2. Collection of Data

Utilizing a Somatom Sensation 16 spiral CT device (Siemens, Munich, Germany), a series of axial CT images were acquired from the superior border of the first to the inferior border of the fifth lumbar vertebra in a volunteer. The individual was placed in dorsal recumbency to ensure orthogonality of the body axis to the imaging plane. Each image was preserved in digital imaging and communications in medicine (DICOM) format with a bone structure window featuring an interlayer distance of 0.75 mm, culminating in 377 slices.

### 2.3. Model Establishment

CT-acquired DICOM data were imported into Mimics 21.0 (Materialise, Leuven, Belgium) for segmentation based on grayscale thresholds ranging from 216 to 2944 Hounsfield units to delineate osseous structures, resulting in a 3D reconstruction of the lumbar vertebrae L1-L5. Subsequent refinement was performed in Geomagic Studio (Geomagic, Inc., Research Triangle Park, NC, USA), involving porosity fills and edge smoothing, meticulously preserving the original curvature to retain primary morphological features. The isolated vertebral models were then assembled with an IVD construct in SolidWorks 2021(Dassault Systèmes, Waltham, MA, USA), using anatomical landmarks to extrude endplates from each vertebra’s superior and inferior surfaces. Components representing the nucleus pulposus, annulus fibrosus, and articular cartilage were subsequently modeled on the endplate surfaces. The assembly was meshed using HyperMesh 2019 (Altair Engineering, Inc., Troy, MI, USA). The cancellous bone was modeled using C3D4 tetrahedral solid elements. The external layer of this bone was converted into a 1 mm thick cortical bone shell, defined as S3R three-node shell elements. The nucleus pulposus, endplate, and annulus fibrosus represented C3D8H hexahedral solid elements. Additionally, the ligament was modeled as a ligament as a T3D2 line element, as illustrated in Figure 2. FE analysis was conducted in Abaqus 2021 (Dassault Systèmes Simulia Corp., Johnston, RI, USA), applying physiologically relevant boundary conditions to explore biomechanical responses.

Under the actual kinematic attributes of the lumbar spine, the contact condition between the annulus fibrosus and the nucleus pulposus of the IVD is designated as “bonded” due to their characteristic of remaining unified and inseparable under a normal load, simulating their unified behavior in a physiological state. Concerning the vertebral bodies and facet joints, which may exhibit sliding under specified loading conditions, the contact condition is set to “frictional”, with a friction coefficient of 0.01 [34] to represent the low-resistance sliding attributed to the synovial fluid within the joint spaces. Additionally, to maintain the simulation’s structural cohesion and fidelity, the interfaces between cortical bone and cancellous bone, and between cortical bone and the endplate, are defined as shared nodes. In contrast, the endplate and the IVD are connected via a “tie” constraint. This arrangement denotes that they are virtually coalesced within the model, ensuring the proper transmission and dispersion of forces. Components within the model are characterized as elastomeric materials, with the skeletal framework and discs conceptualized through isotropic elasticity. Empirical investigations have guided the choice of material properties for the model’s bony structures, IVD, and ligamentous components [35,36,37]. To gain a comprehensive understanding of the material parameters of the skeletal framework and the associated soft tissues, one must consult Table 1 [35]. In instances of IVD herniation, histopathological examination predominantly reveals the desiccation of the nucleus pulposus, and breaches in the annulus fibrosus integrity, alongside modifications in the biochemical makeup and permeability of both the annulus fibrosus and the nucleus pulposus [38,39,40]. These alterations markedly diminish the hydration levels of the annulus fibrosus and appreciably escalate the elastic modulus of the IVD when juxtaposed with their normal counterparts. This investigation adopts an IVD herniation model, orchestrated by adjusting the material attributes of the nucleus pulposus and annulus fibrosus [35], as detailed in Table 2. The complete three-dimensional FE model of the lumbar spine consisted of 419,439 total meshes and 147,723 total nodes.

Due to the model being based on the physiological structure of a weightlifter’s lumbar spine, the physiological straightening model cannot be compared with a normal lumbar spine. Based on the anatomical indicators of the human frame and appropriate material properties, this graduate student developed a detailed L1-L5 FE representation of the weightlifter’s specific lumbar spine. The morphological consistency between the designed model and the actual structure was verified by comparison with computed tomography (CT) images to ensure the validity of the model [41].

### 2.4. Boundary and Loading Condition

Lifting an object symmetrically with both hands is common, involving the coordinated action of skeletal joints, muscles, and other body parts. The internal mechanical behavior of the human body is complex. When using simulation techniques to model real-world situations, it is necessary to approximate actual stress models. This study adopts a simplified mechanical model, defining the trunk flexion angle as 45° for symmetrical lifting and treating the lumbar spine as a rigid body. According to the theory of static mechanical equilibrium, the moment exerted by the erector spinae muscles to maintain lumbar flexion is calculated. This approach is based on the mathematical model developed by Bao et al. [42], which estimates the lumbar forward bending moment in athletes preparing to lift a barbell. The estimation model is shown in Figure 3.

Taking the lumbosacral region as the pivot point, N represents the force acting on the sacral region, N1, and N2 the counterweight; F is the strength of the erector spinae to maintain spine flexion; F1 is the total mass of the trunk; and F2 is the cumulative weight of the head, neck, arms, and the lifted object. S is the distance of the erector spinae from the spine, while S1 and S2 are the moment arms of F1 and F2, respectively [43].

According to the principles of theoretical mechanics’ equilibrium theory, the entirety of the force system must satisfy conditions of force balance across horizontal and vertical axes as well as rotational equilibrium. The equation is as follows:(1)∑M=0
(2)$$F\times S=F_{1}\times S_{1}+F_{2}\times S_{2}$$

The torso accounts for 43% of total body weight. In contrast, the combined weight of the head, neck, and arm is 17%, and the L-arm S from the erector spinae to the lumbar spine is about 5 cm [42]. The distance from the human head to the sacrum is considered a straight-line distance during bending and lifting. The length of the head and neck accounts for 19% of the height and 29% of the torso [44], the S1 is about 35 cm, and the S2 is about 57 cm. We set the weight of the lifted object as 15 kg, 20 kg, and 25 kg to calculate flexion moment as 26.12 Nm, 28.97 Nm, and 30.68 Nm, respectively.

According to Bao et al. [42], body mass above L1 is estimated concerning Chinese human body parameter values. This parameter is obtained from many Chinese human body samples investigated by Zheng et al. [45]. Based on the weightlifter’s height and weight, the estimated weight on L1 is about 28.6 kg.

We fully restrain the lower surface of the L5 vertebra. The moment exerted by the erector spinal muscles to maintain lumbar spine flexion and the body weight acting on the upper surface of the L1 vertebral body are applied as loading conditions (Qing-Hua and Chun-yu) to simulate the static action of bending and lifting an object. Flexion moment and vertical loads are applied to the upper surface of L1, as shown in Figure 4.

## 3. Results

### 3.1. Whole Vertebrae Deformation during Weight Lifting

Under the three kinds of load, the displacement and range of motion (ROM) of the lumbar spine are similar; flexion shows an upward trend with the increase in load, and the extension almost does not change. In addition, moving more to the right and bending slightly to the right, little changes as the load increases. At the same time, the left–right rotation is small, with minimal change as the load increases, and the right rotation is slightly more than the left rotation. Due to the changing trends being similar, we only present the displacement and rotation cloud diagrams when simulating lifting a 15 kg object, as shown in Figure 5. For data under three loads, refer to Table 3.

### 3.2. Distribution of Biomechanical Forces in FE Modeling of the Lumbar Spine

Under three different loading conditions, the stress distribution trends of the lumbar spine are similar, primarily concentrating within the vertebral bodies, and, with increasing load, stress values escalate. In the overall lumbar spine, the principal stresses in the lumbar spine are concentrated at the upper and lower anterior margins of the vertebral bodies, with the L5 vertebral body’s anterior aspect experiencing the highest concentration of stress. The maximum stresses are localized at the lower anterior edge of the L5 vertebral body, measuring 175.50 MPa, 184.30 MPa, and 189.60 MPa, respectively. Stress concentrations are also noted near the junctions between vertebral bodies and facet joints, particularly at posterior junctions and at ligament and spinous process interfaces, where the posterior structures of the L5 vertebra bear the most stress. Within the vertebral bodies, stress primarily concentrates at the upper and lower anterior edges, diffusing towards the middle portions, with the highest stress observed at the upper edge of the L5 vertebral body. Additionally, significant stress concentrations are evident at the posterior margins of the L1 and L5 vertebral bodies. In the vertebral regions, stress is predominantly concentrated at the upper and lower anterior edges, spreading towards the central regions, with the highest concentration observed at the upper anterior edge of the L5 vertebral body, while significant stress concentrations are also evident at the posterior edges of the L1 and L5 vertebral bodies. In the IVD regions, the most extensive stress concentrations occur in the degenerated L4-L5 IVD, including its anterior upper and lower edges, and the posterior lower edge. The anterior foremost edge and outer surface of the L3-L4 IVD also show notable stress concentrations, while stress distributions in the L1-L2 and L2-L3 regions are comparatively more diffuse and localized. Overall, the lumbar spine exhibits uneven stress distributions among the vertebral bodies and IVDs, as shown in Figure 6.

## 4. Discussion

FE analysis is particularly conducive to studying abnormal bony structures, avoiding the need for implantable structures. This non-invasive method allows for the detailed study of internal stress, predicting potential failure points and areas of high stress, thus aiding in designing preventive measures and rehabilitation strategies [46,47]. Our study involves constructing a three-dimensional FE model based on the case of a professional weightlifter with a damaged lumbar spine. The aim is to predict the biomechanical response of the lumbar spine under symmetric bending and lifting activities by applying external loads, thereby reflecting the ROM and internal stress in the lumbar spine during routine symmetric lifting of everyday materials. In the context of kinematics, our results indicate that under anterior bending moments and vertical loading, the lumbar flexion angle of the weightlifter’s lumbar spine increases with the load. Simultaneously, there is a slight left–right rotation and bending, with minor rightward bending and rotation overall. As the load increases, rotation and flexion remain almost unchanged. The segmental structure of the lumbar spine undergoes Euler rotations under any movement, but the lumbar spine’s physiological structure and pathological conditions influence its kinematics [48]. The mechanisms in this area are complex, and current research is ongoing. Due to the fixed motion simulation in this study, there are no completely equivalent experimental simulations in previous reports. We compared our results with the kinematic study conclusions of Chowdhury et al. [49] on normal lumbar spine during symmetrical bending and lifting. We found several differences. First, the load applied in our study was greater than their study’s variables, but the lumbar flexion angle in our weightlifter seemed smaller than in normal individuals. Additionally, lumbar rotation and lateral bending were almost non-existent and did not change significantly with increased load. Secondly, lumbar rotation and lateral bending were much smaller than normal individuals, whose variability was more pronounced with increased load. This seemed to confirm that the weightlifter’s range of motion was limited. This could be due to the lumbar spine exhibiting reduced curvature in the sagittal plane, leading to decreased buffering capacity, tighter nesting of the vertebral spinous processes and facet joints, and increased lumbar stiffness. This is consistent with flexion stiffness in the low protrusion model of the lumbar spine constructed by Naserkhaki et al. [50]. The common denominator of these two personalized lumbar spine models is that the sagittal curvature of the lumbar spine is smaller than that of normal people. We also simulated IVD degeneration. Ohlmann et al. [51]’s study using FE analysis on the impact of degenerative discs on lumbar mobility found that increased disc degeneration significantly reduces coupled movements of the lumbar spine, affecting vertebral torsion. Cai et al. [52] found that the lumbar ROM would be reduced with the deepening of the degenerated IVD. Many researchers have also confirmed this [53,54]. It is crucial to note that bony structures and lumbar pathologies influence each other, especially since this weightlifter had a history of heavy bending and lifting exercises before falling ill. Therefore, the internal structural mechanism is complex. In our study, the load was much smaller than their usual practice load, but the cumulative effect of small loads seems to push the lumbar spine to the limits of its range of motion in left–right rotation and bending. This suggests that similar patients should reduce bending and lifting tasks, as such actions will likely cause vertebral compression and fractures.

Regarding lumbar spine stress, the overall stress distribution across the lumbar vertebrae is uneven and relatively dispersed. The anterior surface stress of the L5 vertebra is significantly greater than other vertebrae, concentrated at the upper and lower edges of the vertebral body. Concurrently, several stress concentrations are observed in the posterior structures of the L5 vertebra, which may be related to the degeneration of the L4-L5 IVD, hindering effective stress transmission [55,56]. Other stress concentrations are primarily located at the joint interfaces. During symmetrical flexion movements, normal lumbar spine stress typically concentrates on the vertebral bodies [57,58], with rare occurrences of uneven stress distribution. The posterior joints play a protective and connective role in lumbar spine movements; however, stress is also observed in the posterior vertebral structures in this weightlifter, and flexion moment increases the risk of joint fractures. The range of stress in degenerated IVDs is broader [59], similar to normal lumbar discs, with stress concentrating at the disc margins during movements [60]. Naserkhaki et al. [50]’s study also noted that, while different types of lordosis affect the mechanics of flexion and extension, they exhibit similar load sharing when it comes to disc compression. This suggests that, although the geometry and degree of lordosis of the spine structure can alter the spine components’ mechanical response and load distribution, the increased degree of disc compression is similar across spine types. Notably, this weightlifter shows significant stress concentrations at both the anterior and posterior upper and lower edges of the L5 IVD. This suggests potential risk factors during activities such as bending and lifting, which could exacerbate disc degeneration. Previously reported [61] indicates a close interaction between disc degeneration and skeletal abnormalities. Disc degeneration exacerbates the pathological process of skeletal abnormalities, and skeletal abnormalities further accelerate disc degeneration by altering mechanical transmission paths, leading to excessive stress in certain areas [62]. The influence of bony structural abnormalities and lumbar spine pathologies on stress responses is highly complex. This weightlifter’s lumbar spine exhibits a wider overall stress range than normal, likely due to a physiological vertical alignment markedly different from the typical lordotic curvature of normal lumbar spines. This increases overall stiffness and a reduced ability to disperse stress, leading to irregular stress concentration zones. For this weightlifter, it is crucial to reduce the stress concentrations caused by anisotropic lumbar spine loads, prevent the progression of degenerative changes, and implement protective measures.

Instances of lumbar spine structural abnormalities are notably frequent among weightlifters, encompassing conditions such as back pain and impairments in performing daily tasks. Previous studies have documented a myriad of osseous anomalies and early degeneration of IVD among these athletes. The impact of symmetric loading in weightlifting on athletes varies [63,64,65]; some athletes exhibit an increase in lumbar curvature, which might be a beneficial adaptation to enhancing their competitive capabilities [5]. In contrast, adverse reactions similar to those observed in this weightlifter arise, although most research focuses more on restoring athletes’ competitive abilities rather than their capability to perform everyday activities [5]. Symmetric loading exercises hold potential applications across various domains, including daily activities and rehabilitation efforts, resulting in a holistic mechanical response from the body. This study acknowledges certain limitations, such as our modeling of the IVD’s physical structure, which inadvertently increases the overall stiffness of the lumbar spine, a scenario that may differ from reality in stiffness. Furthermore, the degree of disc degeneration is challenging to simulate realistically, given our reliance on materials from previous studies. Moreover, excluding a muscle model and opting instead for theoretical calculations to simulate conditions marks another limitation. Lastly, this study is carried out within a single-case research framework. Due to the inherent complexity of the finite element (FE) model, single-case design is frequently used in current FE investigations [66,67]. However, it is crucial to recognize that genetic variability exists among individuals, which may lead to concerns about the generalizability of the results obtained. To address this issue, future research could consider incorporating batch modeling with multiple samples to improve the external validity of the findings. Despite these constraints, employing FE analysis has significantly supported clinical and practical research. Our research advancements, through elucidating the biomechanical and physiological impacts of symmetric loading in weightlifting, offer substantial implications for the prevention and diagnosis of conditions akin to those observed in the subject of this study.

## 5. Conclusions

In our simulation, we examine a unique weightlifter by utilizing the FE method to simulate symmetric loading of the lumbar spine. These findings underscore the impact of abnormal physiological lumbar curvature and degenerative IVDs on the biomechanics of the spine. With our analysis, we demonstrate abnormalities in the biomechanical transfer of the lumbar spine under symmetric weight, suggesting the weightlifter should reduce anisotropic loading while also being mindful of the effects of loaded flexion positions on the posterior joints, IVDs, and vertebrae. Our study provides valuable insights for the rehabilitation and clinical treatment of patients with similar conditions.

## Figures and Tables

**Figure 1 bioengineering-11-00825-f001:**
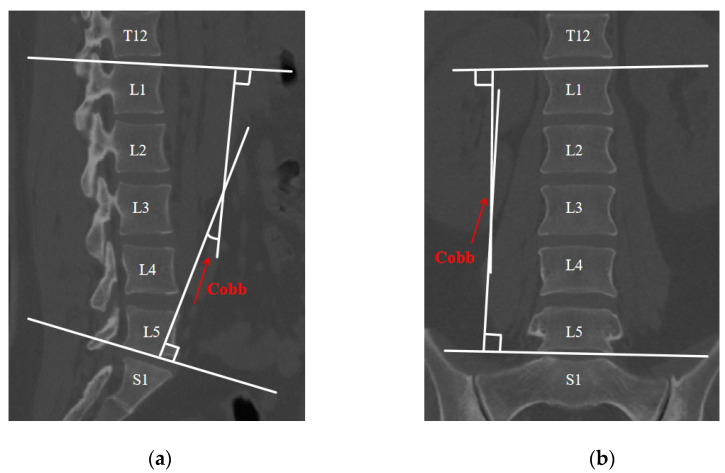
CT image: (**a**) sagittal plane (left); (**b**) coronal plane (right).

**Figure 2 bioengineering-11-00825-f002:**
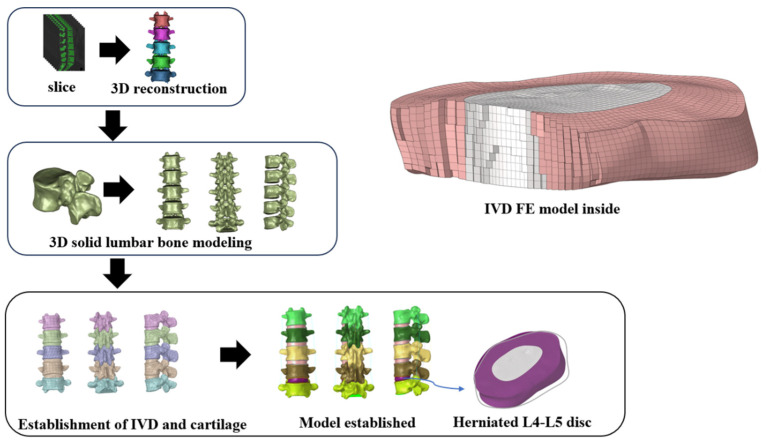
Modeling process.

**Figure 3 bioengineering-11-00825-f003:**
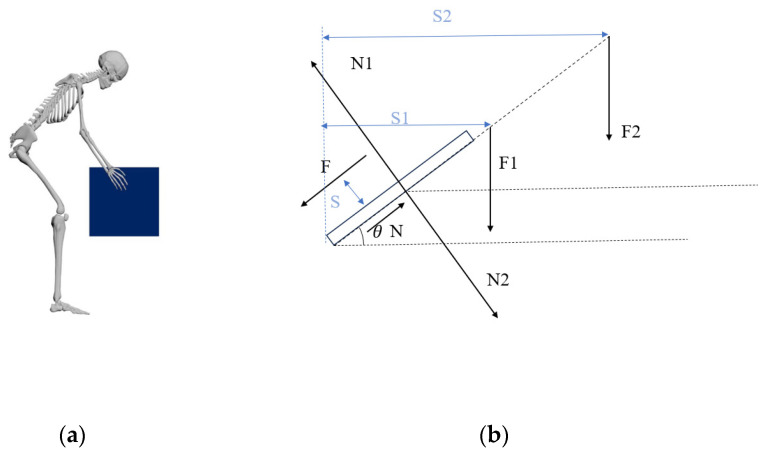
(**a**) Human body symmetrical bending and lifting objects schematic; (**b**) static force analysis of lumbar spine.

**Figure 4 bioengineering-11-00825-f004:**
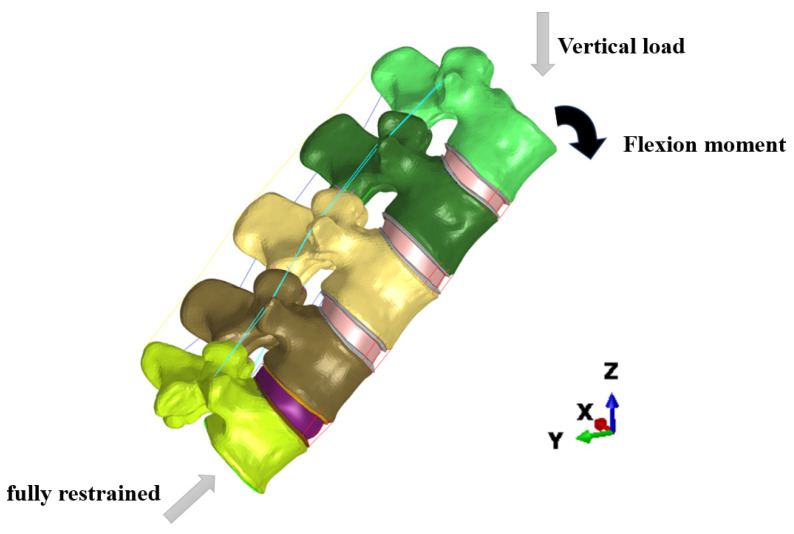
Boundary and loading conditions in FE model.

**Figure 5 bioengineering-11-00825-f005:**
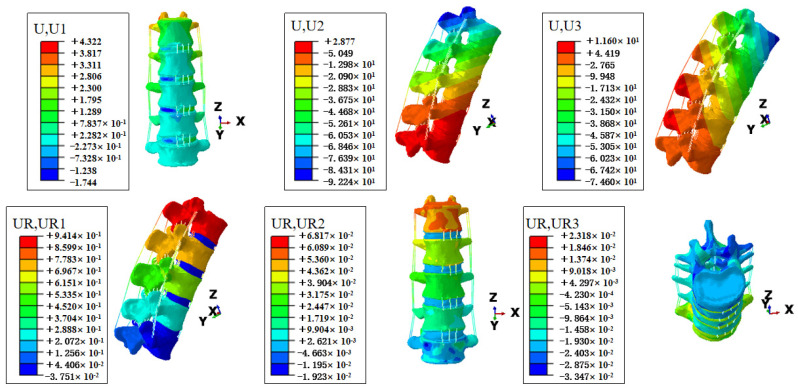
Displacement cloud diagram and rotation cloud diagram when simulating lifting a 15 kg object. In Abaqus, U1, U2, and U3 represent displacements along the X, Y, and Z axes, respectively. The X-axis represents left-right displacement, the Y-axis represents front-back displacement, and the Z-axis represents up-down displacement. UR1, UR2, and UR3 represent rotations around the X, Y, and Z axes in the local coordinate system. Rotation around the X-axis indicates flexion and extension; rotation around the Y-axis indicates left-right bending; and rotation around the Z-axis indicates left-right rotation.

**Figure 6 bioengineering-11-00825-f006:**
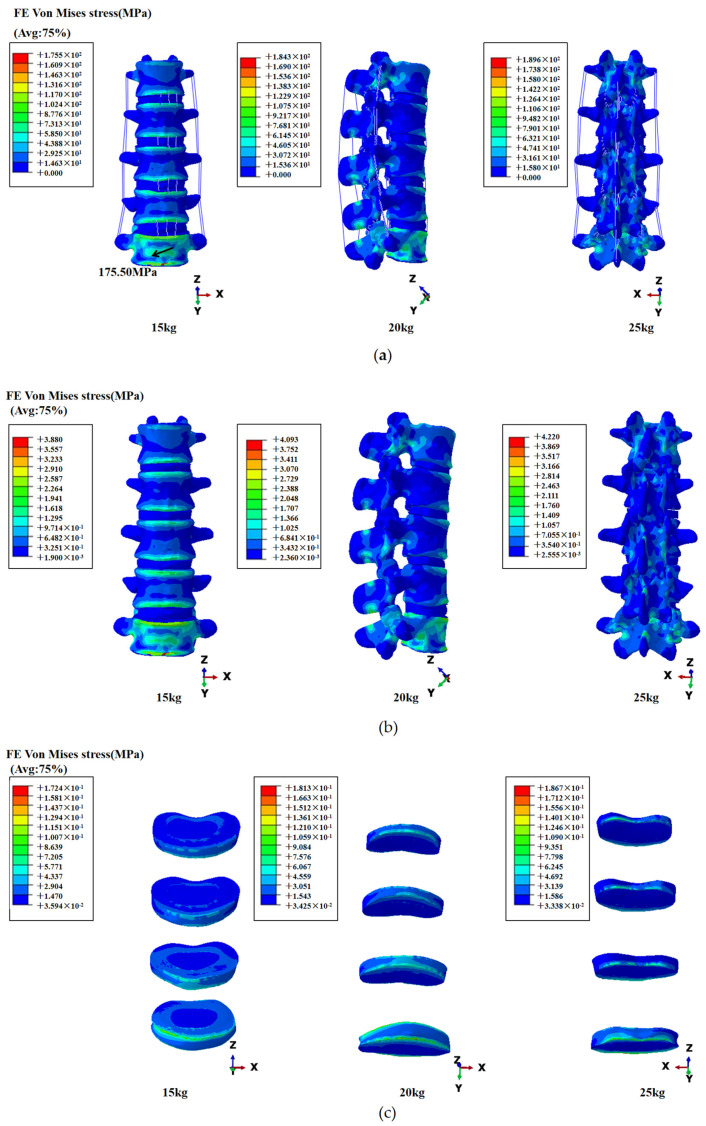
(**a**) Global stress distribution in the lumbar spine; (**b**) vertebral stress distribution; (**c**) IVD stress distribution. From top to bottom are L1-L2IVD, L2-L3IVD, L3-L4IVD, and L4-L5IVD.

**Table 1 bioengineering-11-00825-t001:** Material parameters of the FE model.

	Young’s Modulus (MPa)	Mesh Size (mm)	Poisson Ratio
Cortical bone	12,000.00	1.5	0.30
Cancellous bone	100.00	1.5	0.20
Endplate	25.00	0.8	0.40
Annulus fibrosis	1.00	0.8	0.49
Nucleus pulposus	4.20	0.8	0.45
Anterior longitudinal ligament	7.80	/	0.30
Posterior longitudinal ligament	10.00	/	0.30
Ligamentum flavum	15.00	/	0.30
Interspinous ligament	10.00	/	0.30
Transverse ligament	7.50	/	0.30
Intertransverse ligament	10.00	/	0.30
Supraspinous ligament	8.00	/	0.30

**Table 2 bioengineering-11-00825-t002:** Material setting of IVD herniation between L4-L5.

	Young’s Modulus (MPa)	Poisson Ratio
Annulus fibrosis	1.62	0.40
Nucleus pulposus	10.29	0.40

**Table 3 bioengineering-11-00825-t003:** Displacement and ROM data for three loads.

	15 kg	20 kg	25 kg
	X (+) * X (−) *	X (+) X (−)	X (+) X (−)
UI	4.32 1.74	4.38 1.85	4.41 1.92
UR1 *	53.88 2.29	57.87 2.29	60.15 2.29
	Y (+) Y (−)	Y (+) Y (−)	Y (+) Y (−)
U2	2.88 92.2	3.05 98.2	3.16 101.80
UR2	3.99 1.15	3.99 1.15	3.99 1.15
	Z (+) Z (−)	Z (+) Z (−)	Z (+) Z (−)
U3	11.60 74.60	12.39 79.33	12.86 82.17
UR3	1.14 1.72	1.14 2.29	1.72 2.29

* UR unit: deg; (+) positive direction of the axis; (−) opposite direction of the axis.

## Data Availability

The original contributions presented in this study are included in the article. Further inquiries can be directed to the corresponding authors.

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
