# Peer review of "Biomechanical Study of Symmetric Bending and Lifting Behavior in Weightlifter with Lumbar L4-L5 Disc Herniation and Physiological Straightening Using Finite Element Simulation"

_bioengineering, 2024, doi:10.3390/bioengineering11080825_

Round 1

Reviewer 1 Report

Comments and Suggestions for Authors

The manuscript is about the biomechanical responses of a weightlifter with lumbar straightening and L4-L5 disc herniation during symmetric bending and lifting to optimize training and rehabilitation.

These are some comments to improve the article.

Abstract:

Based on the weightlifter's CT data, an FE lumbar spine model (L1-L5) was established. Please write CT in full before using the abbreviation.

The model includes normal IVDs, vertebral endplates, ligaments, and a degenerated L4-L5 disc. Please write IVD in full before using the abbreviation.

Materials and Methods:

Page 3, line 117. The subject of this investigation is a professional weightlifter, aged 21 years, with a stature of 169 cm and a body mass of 71 kg. Please use common Scientific term e.g. height of 169 cm, body weight of 71kg.

Page 3, line 120. exerting pressure on adjacent nerve roots, and evaluated the lum-119 bar segmental curvature using the COBB angle. What is COBB angle? Please write COBB in full before using the abbreviation.

Page 3, line 122. CT images, as shown in Figure 1, and a comprehensive assessment of symptomatic 122 manifestations, including pronounced lower back and leg pain… Please write CT in full before using the abbreviation.

Page 4, line 135. Each image was preserved in DICOM format with a bone structure window featuring an interlayer distance of 0.75 mm, culminating in 377 slices. Please write DICOM in full before using the abbreviation.

Page 4, line 138. CT-acquired DICOM data were imported into Mimics 21.0 (Materialise, Leuven, Belgium) for segmentation based on grayscale thresholds to delineate osseous structures, resulting in a 3D reconstruction of the lumbar vertebrae L1-L5. What is the value of the grayscale thresholds?

Page 6, line 215. F1is the total mass of the trunk, and F2 is the cumulative weight… please put a space between F1 and is…..

Page 6, line 224-231. Please put a space between the value and unit e.g. 5 cm, 35 cm, 57 cm, 15 kg, 20 kg, 25 kg, 26.12 Nm, 28.97 Nm, 30.68 Nm, etc.

Results:

Page 7 onwards. Paragraphs are not justified.

Discussion:

Page 10, line 323. Rohlmann et al.’ 's [52] study using FE….  Why two ‘ ‘?

References:

Line 508. Why all capital letter for HUANG?

Line 530. Why all capital letter for ZHENG?

Comments on the Quality of English Language

Minor editing of English language required

Author Response

Thank you very much for your review of the manuscript. We are grateful for your positive feedback. Regarding the aspects that need improvement, we appreciate your constructive criticism and we have studied comments carefully and made point-to-point corrections, which we hope the paper is now suitable for publication. Meanwhile, we have asked a native writer to improve the language of our study. The revised portion is highlighted in red and please see the attachment, in blue, for a point-by-point response to your comments and concerns.

Reviewer 2 Report

Comments and Suggestions for Authors

From a methodological point of view, the work presented presents a serious problem: it is based on just one participant. In this sense, I think it should be based on a larger number of participants, because there is always genetic variability between individuals, which in this case has not been safeguarded. In this sense, I suggest that the authors increase the number of participants before it is accepted.

Author Response

Thank you very much for your attention and comments concerning our manuscript. We have studied comments carefully and made point-to-point corrections, and the revised portion is highlighted in red in the paper. Please see the attachment, in blue, for a point-by-point response to your comments and concerns.

Round 2

Reviewer 1 Report

Comments and Suggestions for Authors

The authors have adequately addressed the comments in the revised version of the manuscript. Therefore, I have no further comments.

Comments on the Quality of English Language

Minor editing of English language required